# Direct neural pathways convey distinct visual information to *Drosophila* mushroom bodies

**Katrin Vogt**[1,2†‡], **Yoshinori Aso**[3†], **Toshihide Hige**[4§], **Stephan Knapek**[1], **Toshiharu Ichinose**[1,2], **Anja B Friedrich**[1], **Glenn C Turner**[4§], **Gerald M Rubin**[3*], **Hiromu Tanimoto**[1,2*]

[1]Max-Planck Institut für Neurobiologie, Martinsried, Germany; [2]Tohoku University Graduate School of Life Sciences, Sendai, Japan; [3]Janelia Research Campus, Howard Hughes Medical Institute, Ashburn, United States; [4]Cold Spring Harbor Laboratory, Cold Spring Harbor, United States

**Abstract** Previously, we demonstrated that visual and olfactory associative memories of *Drosophila* share mushroom body (MB) circuits (*Vogt et al., 2014*). Unlike for odor representation, the MB circuit for visual information has not been characterized. Here, we show that a small subset of MB Kenyon cells (KCs) selectively responds to visual but not olfactory stimulation. The dendrites of these atypical KCs form a ventral accessory calyx (vAC), distinct from the main calyx that receives olfactory input. We identified two types of visual projection neurons (VPNs) directly connecting the optic lobes and the vAC. Strikingly, these VPNs are differentially required for visual memories of color and brightness. The segregation of visual and olfactory domains in the MB allows independent processing of distinct sensory memories and may be a conserved form of sensory representations among insects.

**\*For correspondence:** rubing@ janelia.hhmi.org (GMR); hiromut@ m.tohoku.ac.jp (HT)

[†]These authors contributed equally to this work

**Present address:** [‡]Harvard University, Center for Brain Science, Cambridge, United States; [§]Janelia Research Campus, Howard Hughes Medical Institute, Ashburn, United States

**Competing interests:** The authors declare that no competing interests exist.

## Introduction

Rewarding or punitive stimuli modulate behavioral responses to sensory stimuli. In insects, such associative modulation takes place in the mushroom body (MB) (*Heisenberg, 2003*). In the fruit fly, the MB receives distinct dopaminergic inputs that signal reward and punishment, and this valence circuit in the MB is shared for associative memories of different modalities: olfaction, gustation, and vision (*Masek et al., 2015*; *Aso et al., 2012*; *Liu et al., 2012*; *Perisse et al., 2013*). The role of MB output during memory acquisition and testing is similar in visual and olfactory memories (*Vogt et al., 2014*). However, the required subsets of MB Kenyon cells (KCs) are not the same (*Aso et al., 2014a*; *Vogt et al., 2014*), raising the possibility that memory-relevant visual and olfactory information may be represented by different KC subsets in the MB.

Studies of the neuronal circuits underlying olfactory learning have benefitted from well-characterized neuronal pathways conveying olfactory information to the MB (*Turner et al., 2008*; *Butcher et al., 2012*). Beyond visual associative memory, the MB was shown to be important in various vision-guided behavioral tasks (*Liu et al., 1999*; *Brembs, 2009*; *Zhang et al., 2007*). Yet, how visual information is conveyed and represented in the *Drosophila* MB is totally unknown. MBs of hymenoptera receive visual afferents in their calyces (*Ehmer and Gronenberg, 2002*; *Gronenberg and Hölldobler, 1999*; *Paulk and Gronenberg, 2008*), while such direct visual input from the optic lobes to the MBs has never been observed in dipteran insects (*Mu et al., 2012*; *Otsuna and Ito, 2006*). Therefore, indirect, multisynaptic pathways have been proposed to convey the visual input to the *Drosophila* MB (*Farris and Van Dyke, 2015*; *Tanaka et al., 2008*). In this

report, we demonstrate the existence of two visual projection neurons that directly connect the optic lobes to the MB.

## Results

To understand the representation of visual memory in the MB, we blocked different subsets of KCs using split-GAL4 drivers and UAS-shi[ts1] (*Aso, 2014a*) and behaviorally screened the flies for color discrimination memory using an aversive reinforcer (*Figure 1C*). Memory was consistently impaired when we used drivers labelling the γ-lobe neurons, confirming their importance for visual memory (*Figure 1—figure supplement 1A*) (*Vogt et al., 2014*). Strikingly, this screen further suggested a subset of the γ neurons (γd) to be specifically responsible (*Figure 1*, *Figure 1—figure supplement 1A*).

The γd neurons are embryonic born KCs consisting of ca. 75 cells in the adult brain (*Butcher et al., 2012*; *Aso et al., 2009*; *Aso et al., 2014b*). Two split-GAL4 drivers *MB607B* and *MB419B* showed strong expression in the γd neurons but had no detectable expression in other KCs (*Figure 1A*, *Figure 1—figure supplement 1B*). The blockade of the γd neurons using these lines severely impaired visual memory (*Figure 1D,E*, *Figure 1—figure supplement 1C*). In contrast, olfactory memory of these flies was not significantly affected (*Figure 1F*, *Figure 1—figure supplement 1D*). Given that both visual and olfactory memories were reinforced by the same aversive stimulus - electric shock punishment - the selective γd requirement for visual memory suggests that these KCs represent visual stimuli.

We characterized γd cell morphology using the *MB607B-GAL4* and *MB419B-GAL4* drivers to express axonal and dendritic markers. Their axons run in parallel to those of the other KCs in the peduncle, and project to the dorsomedial tip of the γd lobe (*Figure 1B*, *2A*). The γd neurons are atypical in that their dendrites are highly enriched ventrolaterally outside the main calyx, where olfactory projection neurons (OPNs) terminate (*Figure 1B*, *2B*), and form the ventral accessory calyx (vAC) (*Butcher et al., 2012*; *Aso et al., 2014b*). Nevertheless, the γd neurons are equipped with claw-like dendritic endings forming microglomeruli similar to those in the main calyx (*Figure 2C,D*).

To examine if the γd neurons are tuned to visual stimuli, we measured electrophysiological responses using whole-cell patch-clamp recordings. In line with the specific requirement of γd neurons for visual conditioning, we found spiking responses to blue and green light stimuli in some of these cells (*Figure 2E,G*). Stimulating flies with 5 different odors did not lead to excitatory responses of the γd neurons; olfactory stimulation rather evoked slow inhibitory responses, implying the existence of feedforward inhibition through other odor-responsive KC populations (*Figure 2E,G*). This response profile is in sharp contrast to what we observed with typical KCs (e.g. α/β neurons), many of which responded to odors but none to visual stimuli (*Figure 2F,G*, [*Turner et al., 2008*]). Thus, there is clear modality segregation between γd and α/β neurons.

The selective tuning of the γd neurons to visual stimuli prompted us to ask if *Drosophila* MBs receive direct visual input from the optic lobes in the γd dendrites. Performing an anatomical screen of a GAL4 driver collection (*Jenett et al., 2012*), we identified two types of neurons with arbors in the optic lobes and projections in the area of the vAC, making them candidates for visual projection neurons (VPNs) to the MB (*Figure 3—figure supplement 1*). We named these two types of VPNs VPN-MB1 and VPN-MB2.

To precisely map the morphology of these VPNs, we generated split-GAL4 drivers, *MB425B* and *MB334C*. These drivers have strong expression in VPN-MB1 and VPN-MB2 with little other expression (*Figure 3A,C*). *MB425B-GAL4* has predominant expression in VPN-MB1 (*Figure 3B*) projecting from the medulla to the vAC with many cell bodies on the anterior surface of the optic lobe. In contrast, *MB334C-GAL4* strongly labels 1–3 cells of VPN-MB2 as well as the MB output neurons MBON-α1 (*Figure 3C,D*) (*Aso et al., 2014a*; *2014b*). We found that the majority of VPN-MB1 and VPN-MB2 dendrites are localized to the optic lobes (*Figure 3E,G*). The VPN-MB1 neurons have dendritic arbors in the medulla enriched in layer M8 (*Figure 3F*). They cover a large field of the medulla with a conspicuous elaboration in the ventral half (*Figure 3A,E,F*). Single-cell labeling revealed that each VPN-MB1 cell samples input from approximately 20 optic cartridges in the medulla (*Figure 3B*). Dendrites of a single VPN-MB2 cover a large field of the ventral medulla, arborizing in layer M7 (*Figure 3C,D,G,H*). The output sites of both VPNs are restricted to the lateral protocerebrum,

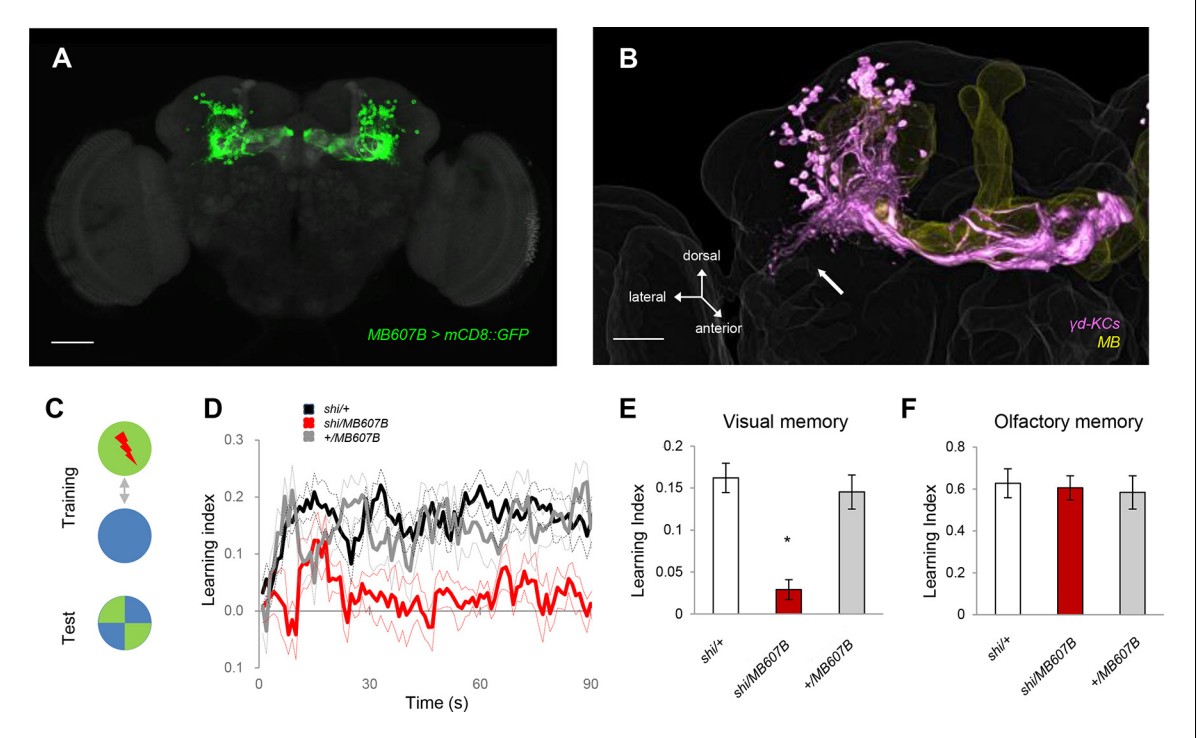

**Figure 1.** The γd KCs are required for visual memory. (A) γd neurons labeled by *MB607B-GAL4.* (B) 3D reconstruction of γd neurons labeled by *MB607B-GAL4* (purple) in the entire MB (yellow). Arrow indicates atypical dendritic protrusion of the γd neurons. Scale bars: 50 μm (A) and 20 μm (B). (C) Schematic diagram of color discrimination learning and test. (D) Average time courses of conditioned color avoidance in the test for flies with the blockade of the γd neurons with *MB607B-GAL4* (red) and the parental controls (black and gray). (E) Pooled conditioned color avoidance. Blocking the γd neurons with *MB607B-GAL4* impairs aversive color discrimination learning (one-way ANOVA, *post-hoc* pairwise comparison, p<0.05; n = 8–12). (F) The same Shi[ts1] blockade of the γd neurons does not impair immediate aversive olfactory memory (one-way ANOVA, *post-hoc* pairwise comparison, p>0,05; n = 9–10). Throughout this study, bars and error bars display mean and SEM, respectively.

The following figure supplement is available for figure 1:

**Figure supplement 1.** Behavioral screen identifies the requirement of the γd KCs in aversive visual conditioning but not olfactory conditioning.

including the vAC, forming pre-synaptic boutons in microglomeruli (*Figure 3E,G*; *Figure 3—figure supplement 2*).

To visualize possible connections between the VPNs and KCs in the vAC, we performed counter-staining of VPNs and the catalytic subunit of PKA that marks KCs (*Wolff and Strausfeld, 2015*) and found that the VPN terminals are enwrapped by the dendrites of the γd neurons (*Figure 3—figure supplement 2*). Differential labeling of KCs and the VPNs was consistent with their proposed connection (*Figure 3—figure supplement 3*). We also detected a GRASP signal between the VPNs and the ventrolateral projection of the γd dendrites (*Figure 3I–L*).

We examined the requirement of the VPNs as well as OPNs in visual memory by blocking their output using the driver lines *MB425B-GAL4, MB334C-GAL4* and *GH146-GAL4* (*Figure 4A–C*). Memory was not significantly altered upon the blockade of the OPNs (*Figure 4A*). In contrast, the blockade of VPN-MB1, but not VPN-MB2, significantly impaired color discrimination memory (*Figure 4B, C*). To further substantiate the results with *MB425B-GAL4*, we employed another driver line (*VT008475*) that labels similar VPNs to VPN-MB1. These VPNs in *VT008475* project to the lateral protocerebrum but do not reach the vAC (*Figure 4—figure supplement 1*). The blockade of these VPNs left color discrimination memory intact (*Figure 4—figure supplement 1*). These results strongly suggest the existence of modality-specific input pathways to the MB and that VPN-MB1 is a key component of the visual learning pathway.

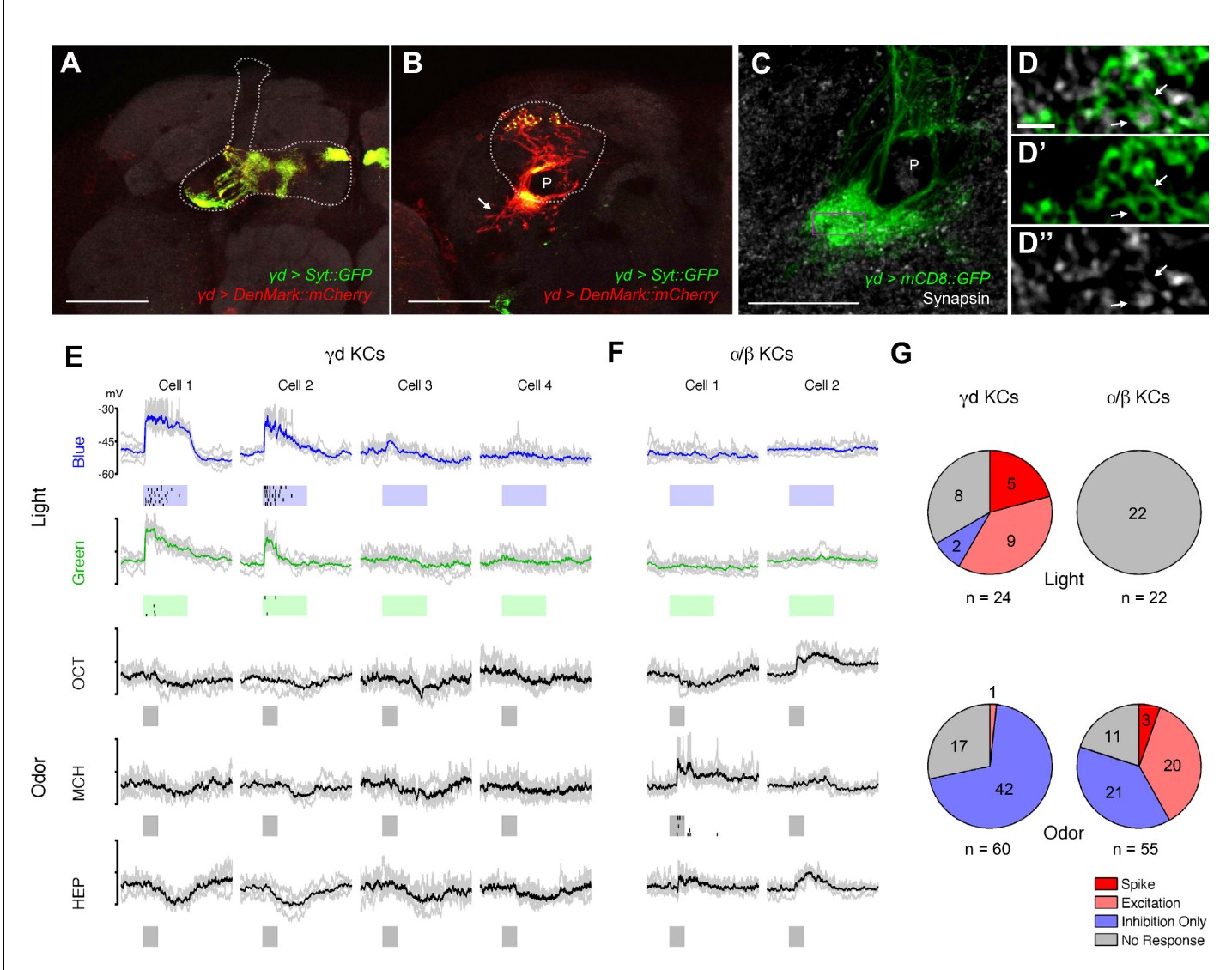

**Figure 2.** γd neurons are atypical KCs and respond to visual stimuli. (**A–B**) Main output and input sites labeled by Syt::GFP (green) and DenMark::mCherry (red) are differentially localized to the dorsal γ lobe and the vAC (arrow). The MB lobe (**A**) and main calyx (**B**) are outlined. P: MB peduncle (**C–D**) The γd dendrites (green) enwrap presynaptic terminals (gray; arrows). A single optical slice of the inset in the projection in **C** is magnified in **D-D''**. P: MB peduncle. Scale bars: 50 µm (**A–B**); 20 µm (**C**); 2 µm (**D-D''**). (**E**) Responses to light and odor stimulation in γd KCs measured with whole-cell current-clamp recordings. Data from four representative neurons are shown (each column corresponds to the data from one cell). Voltage traces of individual trials (gray lines, 5–7 trials) are overlaid with the mean (colored line). Raster plots below the traces represent spikes. Stimulus presentation is indicated below each trace (duration = 1 s). For odors, three of five tested odors are displayed (OCT: 3-octanol; MCH: 4-methylcyclohexanol; HEP: 2-heptanone). (**F**) Responses in two representative α/β KCs. (**G**) Modality segregation by γd (*n* = 12 cells) and α/β KCs (*n* = 11 cells). Each of the pie charts represents 24 (γd) or 22 (α/β) light-cell pairs measured in 6 flies and 60 (γd) or 55 (α/β) odor-cell pairs measured in 3 flies. The distributions of all four response categories are significantly different between γd KCs and α/β KCs with respect to both visual ($p<10^{-5}$, Fisher's exact test) and odor responses ($p<10^{-6}$) See Materials and methods for details.

In our standard learning assay, flies discriminate the chromatic information of blue and green LED stimuli. In addition, flies can learn different light intensities of a single chromatic cue (*Schnaitmann et al., 2013*). We asked if the γd neurons and same VPNs are also required for brightness discrimination learning. We found that while the γd neurons were also necessary for this task, the blockade of VPN-MB1 did not significantly alter performance (*Figure 4D,E*). In contrast, learning was significantly impaired by blockade of VPN-MB2 (*Figure 4F*). Because *MB334C*-GAL4 has expression not just in VPN-MB2 but also in MBON-α1, we tested and ruled out the involvement of MBON-

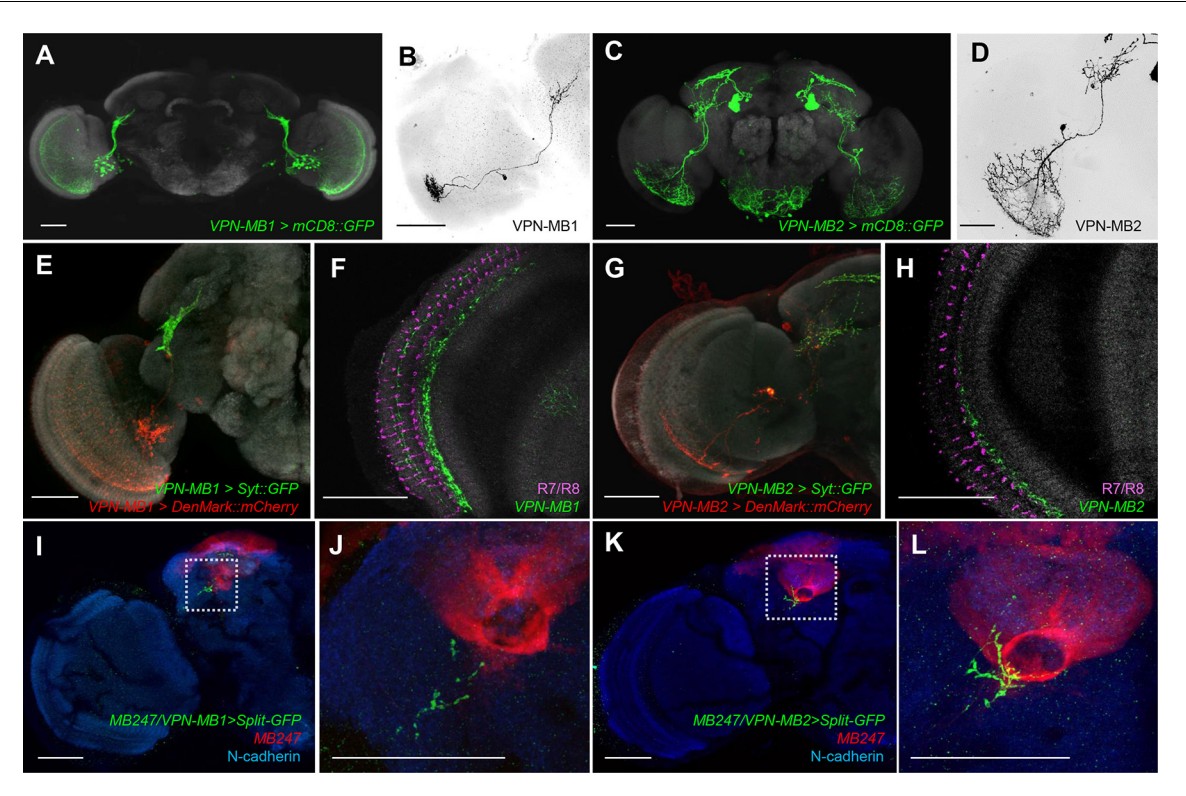

**Figure 3.** VPNs directly convey optic lobe inputs to the MB vAC. (**A**) VPN-MB1 neurons labeled by *MB425B-GAL4.* (**B**) A single VPN-MB1 neuron, generated by heat shock flip out, connects the medulla and the central brain. (**C**) VPN-MB2 neurons labeled by *MB334C-GAL4.* (**D**) VPN-MB2 neurons connect the medulla and the central brain. (**E**) VPN-MB1 has dendrites (DenMark::mCherry, red) in the ventral medulla and presynaptic terminals in the central brain (Syt::GFP, green). (**F**) The dendrites of VPN-MB1 (green) arborize in the M8 layer. (**G**) VPN-MB2 has dendrites (DenMark::mCherry, red) in the ventral medulla and presynaptic terminals in the central brain (Syt::GFP, green). (**H**) The dendrites of VPN-MB2 (green) arborize in the M7 layer. (**I–J**) Reconstituted GFP signals visualize contacts between KCs and VPN-MB1 in the vAC. (**K–L**) Reconstituted GFP signals visualize contacts between KCs and VPN-MB2 in the vAC. **J** and **L** are magnifications of the insets in **I** and **K**. Scale bars represent 50 μm.

The following figure supplements are available for figure 3:

**Figure supplement 1.** 3D reconstruction of VPNs and γd neurons (purple: *MB419B-GAL4*) registered in a standard brain reveals overlapping processes in the vAC (arrow).

**Figure supplement 2.** VPN axons overlap with KC processes in the vAC labeled with DC0.

**Figure supplement 3.** VPNs connect to the γd vAC.

α1 by showing that its blockade did not significantly impair brightness discrimination learning (*Figure 4—figure supplement 2*). The differential requirements for VPN-MB1 and VPN-MB2 indicate that information about color and intensity of a visual cue is separately conveyed to the MB via distinct VPNs.

## Discussion

Previous studies have shown the importance of MB circuits for visual memories and other visually guided behaviors in *Drosophila* (*Vogt et al., 2014*; *Liu et al., 1999*; *Zhang et al., 2007*); however, the MB had been thought to receive visual input from the optic lobe through an unknown indirect pathway, because direct connections had not been observed (*Farris and Van Dyke, 2015*; *Tanaka et al., 2008*). Our identification of direct pathways reveals similarity between dipteran and

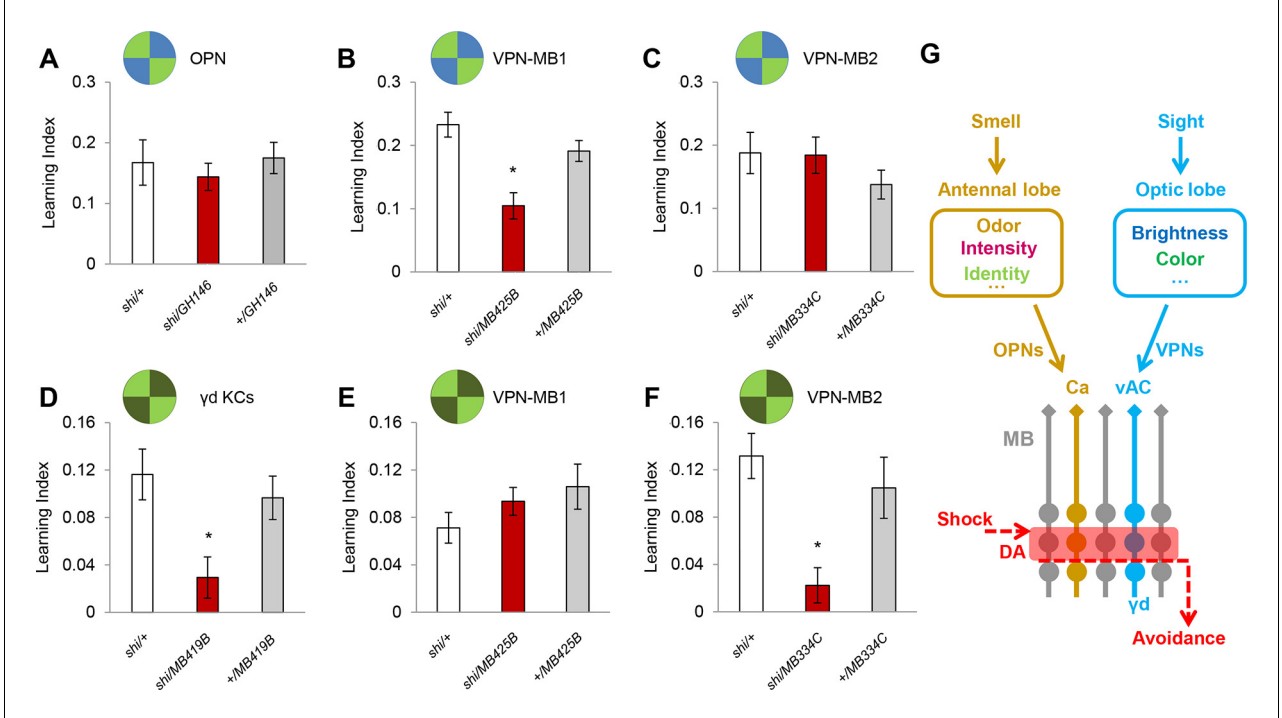

**Figure 4.** VPN-MB1 and VPN-MB2 convey distinct visual features. (A) OPNs labeled by *GH146-GAL4* are not required for visual color conditioning (one-way ANOVA, p>0.05), *n* = 8. (B–C) VPN-MB1 (*MB425B-GAL4*; B), but not VPN-MB2 (*MB334C-GAL4*; C), are required for color discrimination learning (one-way ANOVA, *post-hoc* pairwise comparison, p<0.01). *n* = 9–12. (D) γd neurons labeled by *MB419B-GAL4* are required for green intensity learning (one-way ANOVA, *post-hoc* pairwise comparison, p<0.05), *n* = 9–11; these neurons are also required for color discrimination learning (*Figure 1*). (E–F) In contrast to the requirement in color discrimination learning, the blockade of VPN-MB2 (*MB334C-GAL4*; F), but not VPN-MB1 (*MB425B-GAL4*; E), significantly impaired intensity discrimination learning (one-way ANOVA, *post-hoc* pairwise comparison, p<0.05). *n* = 8–13. (G) Schematic of memory circuits in the MB. Visual and olfactory information is first processed in the optic lobe and antennal lobe, respectively. Components of sensory information (e.g. brightness and color) are separately processed there and conveyed to corresponding KC subtypes in the MB directly through distinct projection neurons Ca: calyx, vAC: ventral accessory calyx. These segregated representations of visual and olfactory information undergo the same dopaminergic (DA) valence modulation to operate acquired behavior (e.g. conditioned avoidance) via shared circuits.

The following figure supplements are available for figure 4:

**Figure supplement 1.** The blockade of similar VPNs without vAC connection does not impair color discrimination learning.

**Figure supplement 2.** The blockade of MBON-a1 does not impair intensity discrimination learning.

hymenopteran circuit design for visual processing, and provides experimental evidence for the behavioral roles of these circuits (*Figure 4G*).

VPN-MB1 and VPN-MB2 receive differential inputs in the medulla and convey distinct visual features - color and brightness - to the MB. The medulla layer M8 where VPN-MB1 arborizes (*Figure 3F*) contains presynaptic terminals of Tm5 neurons that are involved in color vision (*Karuppudurai et al., 2014*; *Gao et al., 2008*). VPN-MB2 is the first *Drosophila* interneuron shown to convey brightness information, and the position of its dendrites suggests that light intensity may be encoded in the medulla layer M7 (*Figure 3H*). Interestingly, the dendrites of VPN-MB1 and VPN-MB2 are preferentially distributed in the ventral half of the medulla, an arrangement consistent with the specialization of the ventral retina for color processing of landmarks during foraging (*Giger and Srinivasan, 1997*; *Kinoshita et al., 2015*; *Wernet et al., 2015*). The target region of the MB-projecting VPNs is separated from other described VPNs in the central brain, and largely segregated from other VPNs mediating innate spectral processing and motion vision (*Mu et al., 2012*; *Otsuna and Ito, 2006*; *Zhang et al., 2013a*). Parallel processing of different visual features with segregated

projections may be a conserved circuit strategy for visual processing across phyla (*Livingstone and Hubel, 1988*).

Our results provide evidence that the *Drosophila* MB represents distinct sensory modalities in different KC subsets whose dendrites are segregated in subdomains of the calyx. Olfactory inputs project to the main calyx and visual stimuli to the vAC, while gustatory stimuli have been recently shown to project to yet another calyx domain (*Kirkhart and Scott, 2015*). In the fly MB, dopamine neurons including those encoding positive and negative valences divide the long axon terminals of KCs into distinct compartments (*Aso et al., 2014a*; *Tanaka et al., 2008*). As KCs send parallel axon fibers, a single dopamine neuron locally modulates the corresponding axonal compartment of multiple KCs (*Hige et al., 2015*; *Cohn et al., 2016*; *Boto et al., 2014*). Distinct KC subsets, γd and γm for example, can therefore share the same dopaminergic valence modulation, even if these KCs are devoted to different sensory modalities (*Figure 4G*) (*Vogt et al., 2014*). These local modulations in turn affect the information conveyed by shared MB output pathways (*Figure 4G*) (*Aso et al., 2014*; *Vogt et al., 2014*). Our results can thus explain the circuit mechanism by which the *Drosophila* MB processes memories of different modalities with shared modulatory and output pathways (*Figure 4G*).

There appears to be a close correlation between the ecological specialization of different insects and the organization of their MB calyces (*Ehmer and Gronenberg, 2002*; *Lin and Strausfeld, 2012*; *Yilmaz et al., 2016*; *Groh et al., 2014*), with the functional subdivision of the calyx reflecting the salient sensory environment. Olfactory processing, which is the dominant sensory modality in *Drosophila* subject to associative modulation, utilizes ∼1800 of the 2000 KCs (*Quinn et al., 1974*). While fruit flies perform a wide range of behaviors driven by visual input, few could be modified by associative learning (*Borst, 2009*), given that fewer KCs subserve visual memory formation. The MB calyces of other insects also possess modality segregation, and different sets of KCs are presumably assigned to each sensory space (*Gronenberg and Hölldobler, 1999*; *Kinoshita et al., 2015*; *Mobbs, 1982*). Our results here are the first to show that individual KCs indeed have unimodal responses. The segregated sensory representation in the MB enables independent formation of different sensory memories, while allowing interaction among distinct KC populations that may underlie complex forms of learning involving multimodal integration (*Zhang et al., 2013b*).

## Materials and methods

### Flies and genetic crosses

Flies were reared at 25°C, at 60% relative humidity under a 12-12-hr light-dark cycle on a standard cornmeal-based diet. Flies were sorted by genotype at least two days prior to experiments. Each behavioral experiment used 30–40 flies of mixed gender under dim red light in a custom-made plastic box, containing a heating element on the bottom and a fan for air circulation.

For behavioral experiments, we used $F_1$ progeny of crosses between females of *w+;;pJFRC100-20x pJFRC100-20XUAS-TTS-Shibire-ts1-p10 in VK00005* (*Pfeiffer et al., 2012*) or WT-females and males of genotypes *MB607B-GAL4* (*Aso et al., 2014b*), *MB419B-GAL4* (*Aso, 2014b*), *GH146-GAL4* (*Stocker et al., 1997*), *MB425B-GAL4*, *MB334C-GAL4*, *MB310C-GAL4* (*Aso et al., 2014b*), *VT008475-GAL4* (VDRC, Vienna, Austria) or Canton-S males. Split-GAL4 lines were generated using described vectors (*Pfeiffer et al., 2010*): *MB425B* carries 28F07-p65ADZp in attP40 and 10E05-ZpGdbd in attP2, *MB334C* carries 52G04-p65ADZp in attP40 and 49F03-ZpGdbd in VK00027. Split-GAL4 lines used in the KC screening (*Figure 1—figure supplement 1*) and for intensity conditioning (*Figure 4—figure supplement 2*) are as described in (*Aso et al., 2014b*). As all transgenes were inserted into the $w^-$ mutant genome, the X chromosomes of the shi$^{ts}$ effector strain was replaced with that of wild-type Canton-S ($w^+$).

### Aversive visual conditioning

We used LEDs to present visual stimuli (green [520 nm] and blue light [465 nm]) from the bottom of the arena as previously described in (*Schnaitmann et al., 2013*). The intensities were controlled by current and calibrated using a luminance meter BM-9 (Topcon Technohouse Corporation) or a PR-655 SpectraScan Spectroradiometer: 19.4 mW/m$^2$ (blue) and 8.58 mW/m$^2$ (green) (*Schnaitmann et al., 2013*). To train flies with different light intensities, blue and green visual cues

were replaced by different intensities of green light (1:10 ratio; 27.8 mW/m$^2$ (bright-green), and 2.77 mW/m$^2$ (dark-green).

For aversive electric shock conditioning, we used an arena with a transparent shock grid as previously described (*Vogt et al., 2014*). During the test phase, the shock arena was video recorded from above with a CMOS camera (Firefly MV, PointGrey, Richmond, Canada) controlled by custom-made software (*Schnaitmann et al., 2010*). Four setups were run in parallel.

Differential conditioning was followed by binary choice without reinforcement (*Vogt et al., 2014*; *Schnaitmann et al., 2010*). Briefly, in a single experiment, approximately 40 flies were introduced into the arena using an aspirator. During a training trial, the entire arena was illuminated with alternating visual stimuli (60 s each) with one stimulus paired with aversive reinforcement. A 1-s electric shock (AC 60 V) was applied 12 times spaced over 60 s during presentation of the punished visual stimulus. Training trials were repeated four times per experiment. In the test, administered 60 s after the end of training, flies were allowed to choose between the two visual stimuli, which were each presented in two diagonally opposed quadrants of the arena. The distribution of the flies was video recorded for 90 s at 1 frame per second. No US was presented in the test period; however, a 1-s shock pulse (90 V) was applied 5 s before the beginning of the test to arouse the flies. Two groups were trained with reciprocal pairings and tested consecutively in the same setup, respectively. The difference in visual stimulus preference between the two groups was then used to calculate a learning index for each video frame. Reinforcement was paired with the first visual stimulus in half of the experiments, and with the second in the remaining experiments, to cancel any effect of order. The whole experimental setup was kept at 33°C for the temperature-induced effect of Shi[ts1].

## Aversive olfactory conditioning

Aversive olfactory conditioning was performed as described in (*Aso et al., 2014a*). A group of about 50 flies in a training tube alternately received octan-3-ol (OCT; Merck, Darmstadt, Germany) and 4-methylcyclohexanol (MCH; Sigma-Aldrich, MO) for 1 min in a constant air stream. OCT and MCH were diluted to 0.6% and 2%, respectively, in paraffin oil (Sigma-Aldrich) and presented in a cup with a diameter of 30 mm. For odor presentation, two of 3/2-way solenoid valves (MFH-3-3/4-S, FESTO, Germany) were used. Each valve was connected to two cups, one of which contains the diluted odor and the other of which contains pure paraffin oil. Four training tubes were connected to the valves. Twelve 1.5 s 90 V electric shocks (DC) were paired with one of the odor presentations. The delivery of electric shocks and the odors was controlled by a custom-made computer program. In the test, the trained flies were allowed to choose between MCH and OCT for 2 min in a T-maze. The odor cups used were identical to the ones for the conditioning. The distribution of the flies was imaged by cameras (FFMV-03M2M, Point Grey), and the preference index was calculated by taking the mean indices of the last 10 s in the 2 min choice. The learning index was then calculated by taking the mean preference of the two reciprocally trained groups. Half of the trained groups received reinforcement together with the first presented odor and the other half with the second odor to cancel the effect of the order of reinforcement. Temperature and humidity were 60% and 33°C, measured with a digital thermo-hydrometer (6011000, Venta Luftwäscher, Germany). Behavioral experiments were performed in dim red light for training and in complete darkness for test.

## Statistics

Statistical analyses were performed with Prism5 software (GraphPad). Groups that did not violate the assumption of normal distribution (Shapiro-Wilk test) and homogeneity of variance (Bartlett's test) were analyzed with parametric statistics: one-sample *t*-test or one-way analysis of variance followed by the planned pairwise multiple comparisons (Bonferroni). Experiments with data that were significantly different from the assumptions above were analyzed with non-parametric tests, such as Mann-Whitney test or Kruskal–Wallis test followed by Dunn's multiple pair-wise comparison (*Figure 1—figure supplement 1*). The significance level of statistical tests was set to 0.05. Only the most conservative statistical result of multiple pairwise comparisons is indicated. Visual conditioning bar graphs show pooled data over total duration of test. Olfactory conditioning bar graphs show pooled data over last 10 s of test.

## Electrophysiology

In vivo whole-cell recordings from KCs were performed as previously reported (*Turner et al., 2008*). Specific cell types were visually targeted using GFP signals, with a 60X water-immersion objective (LUMPlanFl/IR; Olympus) attached to an upright microscope (BX51WI; Olympus). γd and α/β KCs were specifically labeled with GFP by crossing flies bearing UAS-2eGFP (Bloomington) with GAL4 lines, *MB607B* (γd KCs) or *MB008B* (α/β KCs). Adult F1 females were used at 2–3 days after eclosion. The patch pipettes were pulled for a resistance of 6–7 MΩ and filled with pipette solution containing (in mM): L-potassium aspartate, 125; HEPES, 10; EGTA, 1.1; $CaCl_2$, 0.1; Mg-ATP, 4; Na-GTP, 0.5; biocytin hydrazide, 13; with pH adjusted to 7.3 with KOH (265 mOsm). The preparation was continuously perfused with saline containing (in mM): NaCl, 103; KCl, 3; $CaCl_2$, 1.5; $MgCl_2$, 4; $NaHCO_3$, 26; N-tris(hydroxymethyl) methyl-2-aminoethane-sulfonic acid, 5; $NaH_2PO_4$, 1; trehalose, 10; glucose, 10 (pH 7.3 when bubbled with 95% $O_2$ and 5% $CO_2$, 275 mOsm). Whole-cell current-clamp recordings were made using the Axon MultiClamp 700B amplifier (Molecular Devices). Cells were held at around -50 mV by injecting a small hyperpolarizing current, typically less than 2 pA. Signals were low-pass filtered at 5 kHz and digitized at 10 kHz. Spikes were automatically detected by custom-written scripts in Matlab (R2008b, MathWorks) based on their amplitude, after first removing slow membrane potential deflections with bandpass filtering (100 to 1000 Hz); for each recording, we verified the accuracy of this automatic detection algorithm by visual inspection. To detect subthreshold responses, the peak amplitude during a response time window (0 to 0.5 sec after stimulus onset for light stimulation and 0.2 to 3 s for odor stimulation) was calculated using mean voltage traces smoothed with moving average. When amplitudes exceeded 4.2 SDs of the membrane potential fluctuations during baseline (a 1-sec or 3-sec period prior to stimulus onset), we called this an excitatory or inhibitory response. This criterion accurately reflected our visual impression of what was a significant subthreshold response.

For light stimulation, an LED (520 and 468 nm peak wavelength; WP154A4SUREPBGVGAW, Kingbright) was directed at a fly's head from an angle 45 degrees below and directly in front of the animal, at a distance of 11cm. Light intensities were adjusted to match those used in the behavioral experiment by changing the duty cycle of the LED through Arduino Uno (Arduino). All experiments were performed in a semi-dark room. Odors were presented through a custom-built device as described previously (*Honegger et al., 2011*). Saturated vapors of pure odorants were diluted with air at a 1: 20 ratio. Final flow rate was 1 L/min. Odors were presented in a pseudo-random order so that no odor was presented twice in succession. The following five chemicals were used as stimuli: 2-heptanone (CAS# 110-43-0), 3-octanol (589-98-0), 4-methylcyclohexanol (589-91-3), isoamyl acetate (123-92-2) and apple cider vinegar (Richfood).

We observed distinct response profiles of γd KCs and α/β KCs with respect to both light and odor responses (*Figure 2G*). To see if excitatory drive from the two different modalities is segregated between the cell types, we performed similar statistical analysis (Fisher's exact test) after combining the counts of 'Spike' with 'Excitatory', and 'Inhibition Only' with 'No Response' in *Figure 2G*. This analysis also showed highly significant differences both in light ($p<10^{-5}$) and odor responses ($p<10^{-7}$).

## Immunohistochemistry

Adult fly brains were dissected, fixed, and stained using standard protocols (*Aso et al., 2009*). *MB419B-GAL4*, *MB607B-GAL4*, *MB425B-GAL4*, *MB334C-GAL4*, *MB310C-GAL4*, and *VT008475-GAL4* were crossed to *UAS-mCD8::GFP* (*Figure 1*, *Figure 2*, *Figure 3*, *Figure 1—figure supplement 1*, *Figure 3—figure supplement 1*, *Figure 3—figure supplement 2*, *Figure 4—figure supplement 1*, *Figure 4—figure supplement 2*) and stained with anti-GFP antibody (AB) (rabbit anti-GFP polyclonal, Invitrogen, 1:1000) followed by Alexa Fluor 488 (goat anti-rabbit IgG highly cross absorbed, Invitrogen, 1:1000). For neuropil labeling, we used mouse anti-dskl AB (4F3, DSHB, 1:50) followed by Cy3 anti-mouse (Jackson ImmunoResearch, 1:250) or mouse anti-synapsin AB (DSHB, 1:100, (*Klagges et al., 1996*) followed by Cy3 anti-mouse (Dianova, 1:250) or Alexa 633 anti-mouse (Invitrogen, 1:250). *MB419B-GAL4*, *MB425B-GAL4* and *MB334C-GAL4* (*Figure 2D,E, 3C,D*) crossed to UAS-DenMark::mCherry; UAS-Syt::GFP (*Nicolaï et al., 2010*) and double labeling of *w-;UAS-myr-CD8::Cherry,R13F02LexA/CyO;LexAop-GFP/TM2* crossed *to MB425B-GAL4 or MB334C-GAL4* (*Figure 3*, *Figure 3—figure supplement 3*) were stained with anti-GFP AB (rat anti-GFP AB, Chromotek,

1:100) followed by Alexa Fluor 488 (anti-rat AB, Invitrogen, 1:250) and anti-dsRed AB (rabbit anti-dsred, Clontech, 1:100) followed by Alexa Fluor 568 (anti-rabbit AB, Invitrogen, 1:250). DC0-positive neurons (PKA-C1, *Figure 3—figure supplement 1*) were visualized using anti-DC0 AB staining (anti-DC0 rabbit, 1:2000, (*Skoulakis et al., 1993*) followed by Cy3 anti-rabbit (Jackson ImmunoResearch, 1:200). To visualize connections between VPNs and the vAC we crossed w-; mb247-DsRed; mb247-splitGFP11, UAS-splitGFP1-10 (*Pech et al., 2013*) to *MB425B-GAL4* and *MB334C-GAL4*. For visualization of reconstituted GFP (*Figure 3I–J*), we used mouse anti-GFP AB (Clone N86/38, Neuro-Mab, Antibodies Inc., 1:100) followed by goat anti-mouse Alexa 488 (Invitrogen, 1:200). For MB labeling (*Figure 3I–J*), we used rabbit anti-dsRed AB (Clontech, 1: 100) followed by Cy3 anti-rabbit (Jackson ImmunoResearch, 1:200). For neuropil labeling, we used rat anti-N-cadherin staining (anti-N-cad DN-Ex no.8, Developmental Studies Hybridoma Bank, 1:100) followed by Alexa 633 anti-rat (Invitrogen, 1:200). R7/R8 neurons (*Figure 3F,H*) were visualized using mouse anti-MAb24B10 AB (DSHB, (*Zipursky et al., 1984*) followed by Cy3 anti-mouse (Jackson ImmunoResearch, 1:250). Optical sections of whole-mount brains were sampled with a confocal microscope (Olympus FV1000). Images of the confocal stacks were analyzed with the open-source software Fiji (*Schindelin et al., 2012*) and rendered using Fluorender (*Schnaitmann et al., 2010*). We applied the 3D mean filter (r = 2 pixels) followed by deconvolution to high magnification image stacks in *Figure 3D–D*".using Fiji plugins Iterative Deconvolve 3D.

To obtain single-cell flp-out staining, males of the *MB425B-GAL4* were crossed with females of y-w-, *hsp70-flp [X]*; *UAS>CD2 y+>mCD8::GFP/CyO*; *TM2/TM6b* (*Wong et al., 2002*) to obtain $F_1$ progeny carrying GAL4 insertion, hsp70-flp and *UAS>rCD2,y+>mCD8-GFP*. Crosses were raised at 25°C. One to six days before eclosion a mild heat shock was given by placing the vial into a 32°C incubator to remove the FLP-out cassette (rCD2, y+) in a subset of the neurons. The duration of the heat shock was 60–90 min. The eclosed flies were then transferred into a fresh vial and 2- to 5-day-old flies were used for dissection.

## Acknowledgements

We thank Aljoscha Nern for sharing unpublished information on the anatomy of VPNs and critical reading of the manuscript; Irina Sinakevitch for anti-DC0. This work was supported by Max-Planck Gesellschaft (HT), Deutsche Forschungsgemeinschaft (TA 552/5-1 to HT), MEXT/JSPS KAKENHI (26250001, 26120705, 26119503, 15K14307 to HT) and Postdoctoral Fellowship for Research Abroad (TH), the Strategic Research Program for Brain Sciences "Bioinformatics for Brain Sciences" (HT), Naito Foundation (HT), Howard Hughes Medical Institute (GMR and YA), NIH (R01 DC010403-01A1 to GCT), and a Postdoctoral Fellowship from the Uehara Memorial Foundation (TH).

## Additional information

### Funding

| Funder | Grant reference number | Author |
| --- | --- | --- |
| Howard Hughes Medical Institute | | Yoshinori Aso<br>Gerald M Rubin |
| Japan Society for the Promotion of Science | Postdoctoral Fellowship for Research Abroad | Toshihide Hige |
| Uehara Memorial Foundation | Postdoctoral Fellowship | Toshihide Hige |
| NIH Blueprint for Neuroscience Research | R01DC010403-01A1 | Glenn C Turner |
| Max-Planck-Gesellschaft | | Hiromu Tanimoto |
| Deutsche Forschungsgemeinschaft | TA552/5-1 | Hiromu Tanimoto |
| Ministry of Education, Culture, Sports, Science, and Technology | KAKENHI 26120705 | Hiromu Tanimoto |

| Ministry of Education, Culture, Sports, Science, and Technology | The Strategic Research Program for Brain Sciences | Hiromu Tanimoto |
| --- | --- | --- |
| Japan Society for the Promotion of Science | KAKENHI 26250001 | Hiromu Tanimoto |
| Naito Foundation | | Hiromu Tanimoto |
| Ministry of Education, Culture, Sports, Science, and Technology | KAKENHI 26119503 | Hiromu Tanimoto |
| Japan Society for the Promotion of Science | KAKENHI 15K14307 | Hiromu Tanimoto |

The funders had no role in study design, data collection and interpretation, or the decision to submit the work for publication.

### Author contributions
KV, Conception and design, Acquisition of data, Analysis and interpretation of data, Drafting or revising the article; YA, TH, Conception and design, Acquisition of data, Analysis and interpretation of data, Drafting or revising the article, Contributed unpublished essential data or reagents; SK, Acquisition of data, Analysis and interpretation of data; TI, ABF, Acquisition of data, Analysis and interpretation of data, Drafting or revising the article; GCT, Conception and design, Drafting or revising the article; GMR, Conception and design, Analysis and interpretation of data, Drafting or revising the article, Contributed unpublished essential data or reagents; HT, Conception and design, Analysis and interpretation of data, Drafting or revising the article

### Author ORCIDs
Yoshinori Aso, http://orcid.org/0000-0002-2939-1688
Hiromu Tanimoto, http://orcid.org/0000-0001-5880-6064

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
