## [Decision Letter]

Thank you for submitting your work entitled "Direct neural pathways convey distinct visual information to *Drosophila* mushroom bodies" for consideration by *eLife*. Your article has been reviewed by two peer reviewers, and the evaluation has been overseen by Mani Ramaswami as Reviewing Editor and K VijayRaghavan as the Senior Editor.

The reviewers have discussed the reviews with one another and the Reviewing Editor has drafted this decision to help you prepare a revised submission.

In honey bees and other insects, it is well known that the mushroom body receives visual information. In the original Research Article, the authors showed with convincing behavioral genetic analyses that mushroom body γ-lobe Kenyon cells are required for visual memory. In this "Research Advance", the authors use a refined split GAL4 line to show that a subpopulation of about 75 γ-lobe KCs (γd) are required for visual memory. These neurons have dendrites in the region called the ventral accessory calyx (vAC). With whole-cell patch clamp, they found that these γd KCs respond to visual but not olfactory stimulation. Furthermore, they generated new split GAL4 lines that label two classes of visual projection neurons, VPN-MB1 and VPN-MB2. VPN-MB1 and VPN-MB2 neurons both send axons to innervate the vAC region, and have dendrites in the optical medulla M8 and M7 layers, respectively. Interestingly, VPN-MB1 is required for a color discrimination task but not an intensity discrimination task. Conversely, VPN-MB2 is required for an intensity discrimination task but not a color discrimination task. Based on these observations, the authors conclude that VPN-MB1 and VPN-MB2 neurons receive and convey color and brightness information, respectively, to the MB.

The demonstration and identification of direct visual inputs to the mushroom bodies in *Drosophila* is important and novel since these MB neurons modulate contextual visual learning and the identification of key visual projection neurons involved will enable further research to understand how the mushroom bodies control responses to different sensory modalities. Thus, on balance, the works constitutes a material and useful advance on the original article.

There is however some room for improvement in manuscript.

1) Although a Research Advance, it would be valuable to structure this article more as a short stand-alone unit: with an informative Introduction, a Results section focused on this particular study, and a much deeper Discussion that interprets these new findings. Overall, the manuscript could be written such that there would be minimal need for the reader to go back to Vogt et al., 2014. Such a structure could also serve to contextualize this new work and present it from a broad perspective, not limited to the field of *Drosophila* neurogenetics.

2) The split GAL4 lines labeling the VPN-MB1 or the VPN-MB2 neurons also label other neurons in the central brain. Therefore, it would be excellent if additional electrical recordings from these two cell types could be used to demonstrate that they do indeed respond to distinct features of the visual stimulation. This would greatly strengthen the interpretation of the behavioral genetic experiments. However, the reviewers agree that this is an optional but not obligatory experiment.

3) The learning scores are represented as histograms. The authors should consider adding some actual behavioural traces to exemplify how these scores were derived, and how the mutant flies fail to learn, for example in Figure 1. Also in Figure 1: Please provide a schema of the behavioural assay, at the very least as is presented in Figure 4.

4) Figure 1—figure supplement 1: Was olfactory conditioning performed for all of these MB lines, or only for a few? Can this be clarified in the table? Can information for *MB607B* be included in this table?

5) Figure 2: The identified features in panels B-D are not easy to see. Consider especially improving versions of D-D'. The authors should clarify whether the trials are performed in the same cell? And in how many animals? G) Include the value for the proportion of the different reactions observed, and also a number of experimental animals.

6) Figure 4: The model presented in G is quite compelling but still speculative: the authors are suggesting that dopaminergic effects leading to avoidance behaviour are equally applied to different inputs regardless of sensory modality? This warrants further discussion. Also please clarify acronyms in the legend (vAC, Ca).

7) Results presented in Figure 2 need to be statistically tested. For other figures when statistical significance is provided, it would be easier for the readers if the specific statistical method were provided in the legend. However, the reviewers note that the specific tests are stated in the Methods section.

8) Further discussion points:

A) What about the indirect pathways? For a broader audience, perhaps elaborate on what is known about indirect visual pathways to the mushroom bodies? Is there any feedback from the other central brain neuropils?

B) Do the authors think that only brightness and colour is processed in the "learning centre" of the MB and not other visual cues (or have these not been tested)? Is this perhaps linked to earlier finding about contextual visual learning in the MB? This should be explicitly considered.

9) Introduction, first paragraph: instead of have: has.

10) In the subsection “Electrophysiology”, last paragraph: was the saline solution in the fruit fly / around the fruit fly cooled somehow since the LEDs probably produced additional heat that might affect recordings?

11) Please include a table/ supplementary table all of the olfactory learning data for these MB lines, if such are available.

---

## [Author Response]

1) Although a Research Advance, it would be valuable to structure this article more as a short stand-alone unit: with an informative Introduction, a Results section focused on this particular study, and a much deeper Discussion that interprets these new findings. Overall, the manuscript could be written such that there would be minimal need for the reader to go back to Vogt et al., 2014. Such a structure could also serve to contextualize this new work and present it from a broad perspective, not limited to the field of Drosophila neurogenetics.

We revised the text by structuring it into full-fledged Introduction, Results and Discussion sections. Introduction and Discussion now contain sufficient background information as a stand-alone paper.

2) The split GAL4 lines labeling the VPN-MB1 or the VPN-MB2 neurons also label other neurons in the central brain. Therefore, it would be excellent if additional electrical recordings from these two cell types could be used to demonstrate that they do indeed respond to distinct features of the visual stimulation. This would greatly strengthen the interpretation of the behavioral genetic experiments. However, the reviewers agree that this is an optional but not obligatory experiment.

We agree that the Split-GAL4 lines used in this study, although highly specific, label neurons other than the target VPN-MBs. To further substantiate our conclusions, we performed behavioral experiments with new driver lines.

To further substantiate the role of VPN-MB1 in *MB425B*, we identified a GAL4 line *VT8475* that labels similar VPNs that have projections from the medulla to the lateral protocerebrum, but not to the vAC region. The blockade of these neurons did not impair color discrimination learning (Figure 4—figure supplement 1). These new data further support the hypothesis that the specific connection of the medulla and the vAC by VPN-MB1 is required to convey color information to the MB. We added the description of these results in the seventh paragraph of the Results section.

The splitGAL4 line *MB334C*, which we used to manipulate VPN-MB2, also labels a mushroom body output neuron (MBON-α1; Aso et al., 2014). Since MB output is required for visual learning (Vogt et al., 2014), we controlled for the requirement of the MBON-α1 using a SplitGAL4 line specific for MBON-α1 but not VPNs (*MB310C*; Aso et al., 2014) during blockade with shi^ts^. Upon shi blockade with MB310C, we did not find any impairment in intensity learning experiments (Figure 4—figure supplement 2), further supporting the role of VPN-MB2 for conveying brightness information. We added the description of these results in the ninth paragraph of the Results section.

Electrophysiological recording from these cells is technically very difficult due to the positions of their cell bodies. Recording would require developing a new preparation to improve accessibility; this is not feasible within a few months.

*3) The learning scores are represented as histograms. The authors should consider adding some actual behavioural traces to exemplify how these scores were derived, and how the mutant flies fail to learn, for example in Figure 1. Also in Figure 1: Please provide a schema of the behavioural assay, at the very least as is presented in Figure 4.*

For better comprehension of the behavioral protocol of visual learning, we added a schematic drawing of conditioning to Figure 1 (Figure 1). As suggested, we now show the time course of color choice behavior in the test for flies with blocked γd and for the respective parental control flies (Figure 1).

*4) Figure 1—figure supplement 1: Was olfactory conditioning performed for all of these MB lines, or only for a few? Can this be clarified in the table? Can information for MB607B be included in this table?*

Split-GAL4 line *MB607B* was employed only after the initial screen (Figure 1—figure supplement 1). Given the expression pattern as clean as that of *MB419B* in the screen, we chose it as the best secondary line for confirmation of the requirement of the γd neurons for visual learning.

We understand the potential usefulness of comparing the results from the same GAL4 lines in the screens for olfactory and visual learning. We indeed tested all listed drivers for olfactory conditioning, however with a significantly different protocol (e.g. memory retention time) that precluded us from comparing these datasets.

Instead, we tested the Shi blockade with *MB607B* for immediate olfactory memory to further substantiate the differential requirement of the γd KCs. This blockade, as well as with *MB419B*, failed to impair olfactory memory. We included these new data in Figure 1.

*5) Figure 2: The identified features in panels B-D are not easy to see. Consider especially improving versions of D-D'. The authors should clarify whether the trials are performed in the same cell? And in how many animals? G) Include the value for the proportion of the different reactions observed, and also a number of experimental animals.*

We improved the clarity by adjusting the brightness/contrast for Figure 2, and by reselecting the optical section, applying 3D deconvolution for the panels 2D-D”.

In the legend of Figure 2, we now say: “each column corresponds to the data from one cell”.

We added the actual numbers of recordings and animals directly in Figure 2 as well as in the legend.

*6) Figure 4: The model presented in G is quite compelling but still speculative: the authors are suggesting that dopaminergic effects leading to avoidance behaviour are equally applied to different inputs regardless of sensory modality? This warrants further discussion. Also please clarify acronyms in the legend (vAC, Ca).*

Thanks to this comment, we added substantial explanation of the proposed circuit model into the corresponding part of the Discussion. We also clarified the acronyms in the legend.

*7) Results presented in Figure 2 need to be statistically tested. For other figures when statistical significance is provided, it would be easier for the readers if the specific statistical method is provided in the legend. However, the reviewers note that the specific tests are stated in the Methods section.*

We performed statistical analysis of the distribution of visual versus odor-responding cells in the two types of KCs using Fisher’s exact test. We added this to the legend and Materials and methods.

8) Further discussion points:

A) What about the indirect pathways? For a broader audience, perhaps elaborate on what is known about indirect visual pathways to the mushroom bodies? Is there any feedback from the other central brain neuropils?

No direct or indirect visual pathways to the MB had been previously found in *Drosophila*. Dipteran MBs have long been proposed to receive indirect visual input as had been described in the cockroach (Strausfeld and Li, J Comp Neurol 1999), where no direct VPN has been identified either. We provided this discussion in the first paragraph of Discussion.

Although we described that the MB contains many feedback circuits (Aso et al. eLife 2014), it is unknown which one is involved in visual processing. Direct connections from the central complex (or other prominent neuropils) to the MB have never been described.

*B) Do the authors think that only brightness and colour is processed in the "learning centre" of the MB and not other visual cues (or have these not been tested)? Is this perhaps linked to earlier finding about contextual visual learning in the MB? This should be explicitly considered.*

From the rich body of literature, it is clear that color and brightness are not the only visual cues processed in the *Drosophila* MB. A famous example is the information on visual contexts is processed in the MB (Liu et al., Nature 1999), which we mentioned in the text. However, we hesitate to go in to great depth regarding the link between our results and other visual functions of the MB, since differences between the behavioral tasks and experimental conditions across these studies are too large for a direct comparison.

*9) Impact statement, second paragraph: instead of have: has.*

The text has been changed accordingly.

*10) In the subsection “Electrophysiology”, last paragraph: was the saline solution in the fruit fly / around the fruit fly cooled somehow since the LEDs probably produced additional heat which might affect recordings?*

Our visual stimulation was performed with a single small LED, 11cm away from the fly. The stimulation time was only 1 second with 15s pause between trials. Thus, there shouldn’t be any significant temperature effect on the recordings. Furthermore, the saline was continuously perfused over the fly throughout the experiment, which would minimize any fluctuations in temperature.

11) Please include a table/ supplementary table all of the olfactory learning data for these MB lines, if such are available.

See our response to point 4.